

# TAMNR: a network embedding learning algorithm using text attention mechanism

Wei Zhang[1,2,3], Zhonglin Ye[2,3], Haixing Zhao[1,2,3], Jingjing Lin[2,3,4] and Xiaojuan Ma[5]

[1] School of Computer Science, Shaanxi Normal University, Xining, Qinghai, China
[2] School of Computer, Qinghai Normal University, Xining, Qinghai, China
[3] The State Key Laboratory of Tibetan Intelligent Information Processing and Application, Xining, Qinghai, China
[4] Xining Urban Vocational & Technical College, Xining, Qinghai, China
[5] Qinghai Provincial Radio and Television Bureau, Xining, Qinghai, China

Corresponding author
Haixing Zhao, hxzhao@qhnu.edu.cn

## ABSTRACT

Because many existing algorithms are mainly trained based on the structural features of the networks, the results are more inclined to the structural commonality of the networks. These algorithms ignore the rich external information and node attributes (such as node text content, community and labels, *etc.*) that have important implications for network data analysis tasks. Existing network embedding algorithms considering text features usually regard the co-occurrence words in the node's text, or use an induced matrix completion algorithm to factorize the text feature matrix or the network structure feature matrix. Although this kind of algorithm can greatly improve the network embedding performance, they ignore the contribution rate of different co-occurrence words in the node's text. This article proposes a network embedding learning algorithm combining network structure and co-occurrence word features, also incorporating an attention mechanism to model the weight information of the co-occurrence words in the model. This mechanism filters out unimportant words and focuses on important words for learning and training tasks, fully considering the impact of the different co-occurrence words to the model. The proposed network representation algorithm is tested on three open datasets, and the experimental results demonstrate its strong advantages in node classification, visualization analysis, and case analysis tasks.

## INTRODUCTION

Network structure is a common research structure in the objective world, such as social networks, citation networks, or chemical networks (*Newman, 2010*). Many studies, such as image detection, personalized recommendation, or anomaly detection, use these large, complex networks to obtain new knowledge (*Cohen & Havlin, 2014*; *Qi et al., 2018*). With the rapid development of IT technology, social networks, such as WeChat, Weibo, and

Facebook, have now surpassed hundreds of millions of nodes, requiring updated and more complex research of large-scale networks. Network embedding approaches (*Tu et al., 2017a*; *Tu et al., 2017b*; *Zhang, Yin & Zhu, 2018*; *Yang, Xiao & Zhang, 2022*) have emerged to meet this challenge. These approaches avoid tedious feature engineering and combine the original data in the network with other network applications. Network embedding (*Cui et al., 2019*; *Han et al., 2020*) converts components into vector form, such as nodes, edges or subgraphs in the network, and preserves the properties of the processed objects in the network embeddings to the greatest extent. Network embedding also expresses complex relationships of the original network in a more efficient and intuitive way.

Manual methods are usually required to obtain the structural features of the network when constructing feature engineering in traditional network representation models, and to use these featuers to represent the interaction relationships of nodes. These methods are complex, not suitable for automation, and they use different methods for different domains, so they are not universal. Some classical unsupervised network (graph) embedding learning methods use eigenvalues and eigenvectors, and singular values and singular vectors, also known as spectra of the input data matrix and called as the network embedding learning model based on spectral methods (*Kipf & Welling, 2017*). From the perspective of linear algebra, these methods, including multi-dimensional scaling (MDS; *Venna & Kaski, 2006*), isometric mapping (IsoMap; *Tenenbaum, Silva & Langford, 2000*), local linear embedding (LLE; *Roweis & Saul, 2000*), and Laplacian Eigenmaps (LE; *Belkin & Niyogi, 2001*), are usually regarded as dimensionality reduction techniques. LLE assumes that each node is a linear combination of adjacent nodes in the embedding space and uses the distance between the weighted sum vector represented by the neighbouring node and the vector represented by the central node as the loss function, which is obtained by minimizing the loss function. Laplacian Eigenmaps maintains the structure of the network by adding a penalty term to the objective function, and it makes two nodes adjacent as much as possible in the embedded vector space. Graph representations (GraRep; *Cao, Lu & Xu, 2015*) performs singular value decomposition by defining the network topology information of the $k$-order nearest neighbours, and then connects each of the obtained vector representations to reflect the real network structure information. These methods have computational efficiency limitations because they would like to use these simple models, for example they use the $K$-nearest neighbour algorithm (KNN; *Altman, 1992*) to obtain an affinity matrix. The eigenvector solution is used to learn the embedding of network nodes. The time complexity of these model usually reaches $O(|v|3)$ (*Sun, Zhou & Zhao, 2021*). Because these models are calculation heavy and time consuming, it is difficult to apply them to largescale networks.

Therefore, current network embedding learning methods require to reduce computational complexity and improve its performance. Inspired by the Word2Vec algorithm (*Mikolov et al., 2013b*; *Mikolov et al., 2013a*), DeepWalk (*Perozzi, Rami & Skiena, 2014*) obtains the sequence of nodes by random walk, and uses Word2Vec to learn the embedding vector of the nodes. Node2vec (*Grover & Leskovec, 2016*) adopts a random walk strategy with preference, and trades off the probability of occurrence of nodes in the sequence obtained by random walk between breadth-first and depth-first search to maintain high-order proximity. In order to prevent training from easily falling into local

optimum, hierarchical representation (HARP; *Chen et al., 2018*) uses graphical coarsening to aggregate the nodes of the upper layer in the hierarchical structure to create a hierarchical structure of nodes, and divides the nodes and edges of the original network into a series of network graphs with smaller hierarchical structure. HARP then uses DeepWalk or node2vec to obtain high-performance embedding vectors for continuous feature extraction. Adaptive similarity function (AdaSim; *Zhang, Shang & Qiao, 2021*) uses the features obtained by embedding the network based on random walk to introduce the adaptive similarity function, and obtains the node feature representation by optimizing the graph-based objective function. *Dong & Kaeli (2017)* present DNNMark, a GPU benchmark suite that consists of a collection of deep neural network primitives, covering a rich set of GPU computing patterns.

Inspired by the above algorithms, this article proposes a joint network embedding learning algorithm (TAMNR) that combines network structure and node text features, adding a textual attention mechanism to improve network embedding learning. First, the structural features of the network are modeled. Then, the text features are modeled. Because different text features contribute different weights for the model, an attention mechanism is introduced to assign different weights to the words in the nodes, identifying the important words and reducing the weight of unimportant words. This leads to the ability to screen relatively high-quality feature inputs from the text of the network nodes. In the network structure modeling, structure and text modeling are carried out at simultaneously. To verify the effectiveness of the TAMNR model, experiments are conducted on several real-world network datasets and the TAMNR model is compared with some current mainstream node feature learning methods. The results show that the TAMNR model performs well in most cases.

The main contributions of this article are as follows:

1. This article introduces an innovative approach that combines network structure features and network text features with a joint modeling mechanism. The learned network representation vectors contain both the network structure features and the semantic feature between network nodes, enriching the network features that nodes can express.

2. Unlike previous network representation learning methods, the proposed algorithm in this article considers the impact of different words in the node text on the model when embedding the text features of the network nodes. An attention mechanism is used to model the weight information of co-occurrence words in the model, filtering out useless words and focusing on important words for learning and training. This mechanism greatly improves the performance of the proposed network representation learning algorithm.

3. The proposed network representation algorithm is then tested on open datasets, and the experimental results show that the proposed algorithm exhibits stronger advantages in these tasks such as node classification, visualization analysis, and case analysis. The proposed algorithm is also applicable to large-scale network representation learning tasks.

## RELATED WORKS

There are a large number of the network embedding learning models that learn effective low-dimensional representations. For example, DeepWalk obtains a series of node sequences by performing random walk in the graph, then treats each node sequence as a text sentence, which is then used as the input to the Skip-Gram model (*Perozzi, Rami & Skiena, 2014*) resulting in a low-dimensional representation. The LINE (*Tang et al., 2015*) probabilistically models all node pairs of first-order similarity and second-order similarity in the network, and learns node representation by minimizing the Kullback Leibler (KL) distance. The node2vec model improves DeepWalk's random walk strategy. structural deep network embedding (SDNE; *Wang, Cui & Zhu, 2016*) model first uses a deep neural network, which uses the relationships in the first-order neighboring nodes as supervised information of the self-encoding neural network and the second-order neighboring relationships as the unsupervised information of the self-encoding neural network to obtain the embedding vectors. In this way, the local structure and global structure information is preserved, making this method robust to sparse networks. Graph convolutional network (GCN; *Bruna et al., 2018*) proposes a convolutional neural network for non-euclidean network data by encoding the local structure of the network and the features of the nodes to obtain the embedding vector for the nodes. Because DeepWalk actually decomposes a specific feature matrix, Text Associated DeepWalk (TADW; *Yang et al., 2015*) embeds the textual information of nodes into the matrix decomposition procedure. However, the matrix decomposition process consumes time and space, making it difficult to scale to large networks. Context-enhanced network embedding (CENE; *Tu et al., 2017a*; *Tu et al., 2017b*) regards textual content as a special kind of node and then exploits the structural and textual information to learn network embeddings. *Du et al. (2022)* adopts the probabilistic topic model, such as LDA, Word2vec and Glove, to extract text features, and then uses a classifier to automatically identify the topic category based on the obtained text representation vectors.

Neural networks can fully explore the hidden features in data, the attention mechanism can make the neural network to focus on the important features of its input data, and it has been widely used in different fields and types of tasks such as image processing, speech recognition, natural language processing, and network representation learning. The attention mechanism is a general method that does not depend on a specific framework, most attention models currently including the Encoder-Decoder framework (*Ganea, Bécigneul & Hofmann, 2018*; *Liu, Nickel & Kiela, 2019*) adopts the attention to improve the performance. Hyperbolic graph convolutional neural networks (HGCN; *Chami et al., 2019*) applies the attention mechanism to the Encoder-Decoder framework in machine translation, so that the output of the encoder is the weighted sum of each hidden layer in the encoder process. For very long inputs, this model can alleviate the long-distance dependency problem. With the wide application of attention mechanisms, many attention-variant models have emerged to handle more complex tasks. *Luong, Pham & Manning (2015)* designes three functions suitable for different downstream tasks, and proposes global attention and local attention mechanisms. Global attention calculates

the hidden layer vector and considers all the hidden states of the encoder, while local attention only pays attention to a part of the hidden state of the encoder when calculating the hidden layer vector. *Vaswani et al. (2017)* added a scaling factor to the attention weight function for the possible minimal gradient problem of the SoftMax function in the conventional method, speeding up model training. The Self-Attention method (*Cheng, Dong & Lapata, 2016*) computes attention weights by correlating different locations of a single input. Transformer (*Vaswani et al., 2017*) is the first model that completely discards the recurrence of RNNs, the convolution of CNNs, and solely utilizes attention for feature extraction. Graph attention networks (GAT; *Veličković et al., 2018*) uses the Self-Attention method and then calculates the hidden state of each node by paying attention to the neighbor nodes in the network, obtainning the embedding vectors. GeniePath (*Liu et al., 2019*) is a graph neural network model that can learn adaptive sensory paths in a scalable way. In the adaptive path layer, the breadth search functional unit is used to introduce the attention mechanism to learn the weight of the first-order neighborhood nodes in the adaptive routing layer, respectively. A deep search functional unit is used for extracting and filtering the information converged in higher-order neighborhoods. Context Attention Heterogeneous Network Embedding (CAHNE; *Zhuo et al., 2019*) learns context embeddings for nodes by introducing the context node sequence. The attention mechanism is also integrated into the model proposed in this article to better reflect the impact of context nodes on the current node. *Ji & Zhang (2022)* proposes a similarity calculation method combining semantics and the attention mechanism using prefix words. First, the context information is extracted, and then the title word set is obtained. The semantic enhanced representations for two sentences are obtained through the attention mechanism and character splicing. Text-attention Factorization Mechanism (TAFM; *Zhang et al., 2022*) can extract features through text components, text attention, and N-gram text features, mine potential user preferences, and then it uses convolutional automatic encoder to learn higher-level features. There are currently few textual attention models, and there are relatively few works that apply textual attention models to network structure modeling. This article considers the rich attribute information on network nodes and proposes an attention-based network embedding learning model for the structural embedding of the network to obtain a more accurate joint embedding vector.

# ALGORITHM DESIGN

## Problem description

Networks are also called graphs in data science and computer science, and are generally represented by $G = (V, E)$, with $V$ representing the set of all nodes in the network, $E$ representing the set of all edges in the network, and $|V|$ representing the number of nodes in the network. This article uses $R_v \in R^k$ to represent the network embedding vector obtained from training, which is a $k$-dimensional matrix, and each row in the matrix represents a node's $k$-dimensional network embedding vector, where $k$ is expected to be much smaller than $|V|$, and $att$ is the attention parameter.

In real-world network structure data, there is usually a certain degree of similarity between two connected nodes. Given a graph $G = (V, E)$, for any two nodes $V_i$, $V_j$, if there

is an edge between the two nodes, that is $e_{i,j} \in E$, then $V_i$ and $V_j$ are said to be similar to each other. However, many nodes are far apart in terms of topology, but have similarities and simply considering the network structure does not account for these nodes. Therefore, a text-based feature is designed and attention is added for modeling, allowing this network embedding model to incorporate the text features of the nodes. During the network learning process, the network topology and text features of the nodes are preserved, so that the nodes that are close to each other or have similar text features in the network topology. In the subsequent model design procedures, an attention mechanism for text features is added, so that text features with high contributions have a positive impact on network structure modeling, avoiding the negative impact of unimportant features on network structure modeling.

## CBOW (Continuous Bag-of-Words) model using negative sampling

Word2Vec has designed two models: CBOW and Skip-Gram, and provides two optimization methods: negative sampling and hierarchical SoftMax. DeepWalk is a network embedding algorithm proposed for large-scale networks based on the Word2Vec model, and inherits the training model and optimization method provided by Word2Vec. Although the training accuracy of the CBOW adding negative sampling optimization model is slightly lower than that of the Skip-Gram adding hierarchical SoftMax optimization model, the training speed of the former is much faster than that of the latter. Therefore, the negative sampling optimized CBOW algorithm is used in the model proposed in this article to train the embeddings in the model.

For the current node $v$, the context node of node $v$ is $v_{nb}$, node $v$ is a positive sample, other nodes are negative samples, and the negative sample set of node $v$ is $NEG(v)$, $NEG(v) \neq null$. The sampling label of node $v$ is $L(v)$. For $\forall u \in D$, $D$ is the node set. The sampling result of node $v$ by $L(v)$ is defined as follows:

$$L^v(u) = \begin{cases} 1, u = v, \\ 0, u \neq v. \end{cases} \tag{1}$$

with the positive sample of $L^v(u)$ being 1, and the negative sample of $L^v(u)$ being 0.

For a node $v$ and its context node $v_{nb}$, its positive sample is $(v, v_{nb})$, and the goal is to maximize the following probability:

$$g_c(v) = \prod_{v_i \in \{v\} \cup NEG(v)} p(v_i | v_{nb}). \tag{2}$$

And

$$p(v_i | v_{nb}) = \left[ \sigma \left( e_v^{\mathrm{T}} \theta^{v_i} \right) \right]^{L^v(v_i)} \cdot \left[ 1 - \sigma \left( e_v^{\mathrm{T}} \theta^{v_i} \right) \right]^{1 - L^v(v_i)}, \tag{3}$$

where $\sigma(x)$ is the sigmoid function. $e_v$ is the cumulative sum of vectors of each node in $v_{nb}$, and $\theta^{v_i}$ is the vector to be trained for node $v_i$,

Therefore, $g_c(v)$ can be changed based on Eqs. (2) and (3) as follows:

$$g_c(v) = \sigma \left( e_v^{\mathrm{T}} \theta^{v_i} \right) \prod_{v_i \in \{v\} \cup NEG(v)} \left[ 1 - \sigma \left( e_v^{\mathrm{T}} \theta^{v_i} \right) \right]. \tag{4}$$

For the entire node set C, the overall optimization objective function is as follows:

$$G = \prod_{v \in C} g_c(v). \tag{5}$$

The objective function of the CBOW model based on negative sampling is to take the following logarithmic operation:

$$L(v) = \log G = \log \prod_{v \in C} g_c(v) = \sum_{v \in C} \log g_c(v) =$$

$$\sum_{v \in C} \sum_{v_i \in \{v\} \cup NEG(v)} \left\{ L^v(v_i) \cdot \log\left[\sigma\left(e_v^{\mathrm{T}}\theta^{v_i}\right)\right] + \left(1 - L^v(v_i)\right) \cdot \log\left[1 - \sigma\left(e_v^{\mathrm{T}}\theta^{v_i}\right)\right] \right\}. \tag{6}$$

$$L(v, v_i) = \left(1 - L^v(v_i)\right) \cdot \log\left[1 - \sigma\left(e_v^{\mathrm{T}}\theta^{v_i}\right)\right] + L^v(v_i) \cdot \log\left[\sigma\left(e_v^{\mathrm{T}}\theta^{v_i}\right)\right]. \tag{7}$$

Given the function $L(v, v_i)$ about $\theta^{v_i}$, the gradient calculation is:

$$\frac{\partial L(v, v_i)}{\partial \theta^{v_i}} = \frac{\partial}{\partial \theta^{v_i}} \left\{ \left(1 - L^v(v_i)\right) \cdot \log\left[1 - \sigma\left(e_v^{\mathrm{T}}\theta^{v_i}\right)\right] + L^v(v_i) \cdot \log\left[\sigma\left(e_v^{\mathrm{T}}\theta^{v_i}\right)\right] \right\}. \tag{8}$$

Using the derivative function optimization Eq. (8) of $\log \sigma(x)$ and $\log(1 - \sigma(x))$ results in:

$$\frac{\partial L(v, v_i)}{\partial \theta^{v_i}} = L^v(v_i)\left[1 - \sigma\left(e_v^{\mathrm{T}}\theta^{v_i}\right)\right]e_v - \left[1 - L^v(v_i)\right]\sigma\left(e_v^{\mathrm{T}}\theta^{v_i}\right)e_v =$$

$$\left\{ L^v(v_i)\left[1 - \sigma\left(e_v^{\mathrm{T}}\theta^{v_i}\right)\right] - \left[1 - L^v(v_i)\right]\sigma\left(e_v^{\mathrm{T}}\theta^{v_i}\right) \right\}e_v = \left[L^v(v_i) - \sigma\left(e_v^{\mathrm{T}}\theta^{v_i}\right)\right]e_v. \tag{9}$$

In Eq. (6), C is the corpus after random walk of nodes, which is optimized by the stochastic gradient ascent method. This then leads to the update formula of the parameters. The update formula of $\theta^{v_i}$ is:

$$\theta^{v_i} := \theta^{v_i} + \mu\left[L^v(v_i) - \sigma\left(e_v^{\mathrm{T}} \cdot \theta^{v_i}\right)\right]e_v. \tag{10}$$

Considering the gradient of e in $L(v, v_i)$, and using $e_v$ and $\theta^{v_i}$ symmetry, the update formula of the context total node embedding vector v(u) is:

$$v(u) := v(u) + \mu \sum_{v_i \in \{v\} \cup NEG(v)} \left[L^v(v_i) - \sigma\left(e_v^{\mathrm{T}} \cdot \theta^{v_i}\right)\right] \cdot \theta^{v_i} \tag{11}$$

Finally, Eq. (6) can be simplified as follows:

$$L(v) = \sum_{v \in C} \sum_{v_i \in \{v\} \cup NEG(v)} \log p(v_i | v_{nb}). \tag{12}$$

## TAMNR modeling

A simple and efficient joint learning model is needed to meet the requirements of large-scale network embedding learning tasks. This article proposes a joint network representation learning framework based on the textual attention mechanism. The framework consists of two parts: network node relationship modeling and node text relationship modeling. This

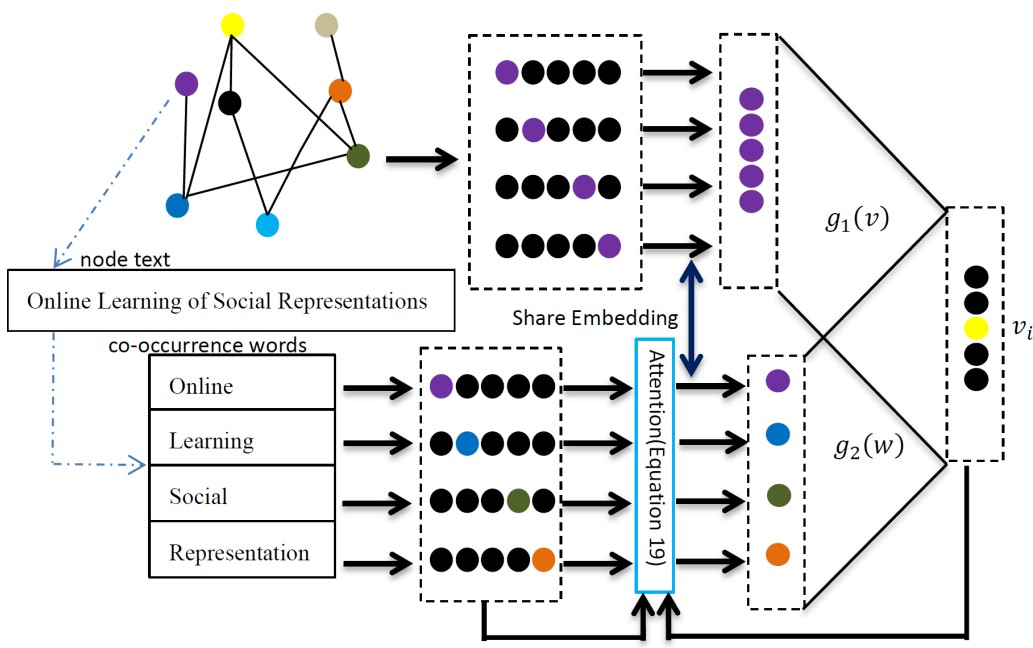

**Figure 1 TAMNR algorithm model.**

model uses text as the input of the relationship model, ensuring that the relationship has the same words in the node text when constructing the node relationship. Through these improvements, the TAMNR model proposed in this article is expected to solve the problem of joint modeling of network structure features and text features, and obtain better quality vector representations. Specific information about the model is shown in Fig. 1.

As shown in Fig. 1, for the current central node $v_i$, the network modeling uses its former two nodes $v_{i-2}$ and $v_{i-1}$, and its next two nodes $v_{i+1}$ and $v_{i+2}$ to predict the probability of the current central node $v_i$ appearing. The network modeling part of the model continuously adjusts the values in the network vector so that the node pairs with connected edges have a closer vector distance to each other, and the node pairs with multi-hop edges or no edges have a farther vector distance. The node text relationship modeling models relationships between node pairs with common text features, and introduces an attention mechanism to give more learning opportunities for node pairs with same words, so that node pairs with the same words have a closer vector distance. Network node relationship modeling and node text feature modeling share a node representation vector in the TAMNR model, allowing these two parts of the model to obtain and exchange feature information through the shared vector, so that the TAMNR algorithm can be used in the modeling learning process. TAMNR can have stronger generalization ability in various tasks by obtaining valuable feature information from neighbor nodes and node text features.

The CBOW model only considers the local contextual information of words, thus failing to effectively capture the quantity and importance of co-occurring words in the text of neighboring nodes. To address this problem, an attention mechanism is added to the TAMNR model to incorporate co-occurring word information, aiming to maximize the

appearance probability of the co-occurring words. The overall objective function proposed is, as follows:

$$L(v) = \sum_{v \in C} \left( \log g_c(v) + \rho \log g_w(v) \right). \tag{13}$$

The left part of the function is the objective function of the relational modeling, the right part is the objective function of the text attention modeling, and $\rho$ is the harmonic coefficient that balances the network structure modeling and the text feature modeling. The goal of this article is to maximize Eq. (13) by using the stochastic gradient ascend method to update each parameter, where $g_c(v)$ and $g_w(v)$ have the same form, and the left term in the above equation is the objective function of CBOW, and its parameters are updated in Eqs. (10) and (11). The network structure model and the text feature model share the same node representation vectors, so they can obtain information from each other through shared representations, enabling the node representation vectors to comprehensively utilize multiple features during training, thereby obtaining higher-quality node representation vectors. The right term of the above equation is the objective function of the text attention model, which is essentially a CBOW model with an attention mechanism added, so the optimization solution is the same as the right term. The right term is defined, as follows:

$$F_1 = \alpha \cdot \sum_{v \in C} \sum_{u \in w_v} \log\left(g_w(u)\right) = \alpha \cdot$$
$$\sum_{v \in C} \sum_{u \in w} \sum_{v_i \in \{v\} \cup NEG(u)} \left\{ L^{v_i}(u) \cdot \log\left[\sigma\left(e_u \cdot \theta^{v_i}\right)\right] + \left(1 - L^{v_i}(u)\right) \log\left[1 - \sigma\left(e_u \cdot \theta^{v_i}\right)\right] \right\}. \tag{14}$$

The left term is defined, as follows:

$$f_1 = L^{v_i}(u) \cdot \log\left[\sigma\left(e_u \cdot \theta^{v_i}\right)\right] + \left(1 - L^{v_i}(u)\right) \log\left[1 - \sigma\left(e_u \cdot \theta^{v_i}\right)\right]. \tag{15}$$

In Eq. (14), the target node is a positive sample, and the remaining nodes are negative samples, and then $f_1$ is used to obtain partial derivatives of $\theta^{v_i}$ and $e_u$ respectively, and the parameter update formula is obtained, as follows:

$$\theta^{v_i} := \theta^{v_i} + \mu \left[ L^{v_i}(u) - \sigma\left(e_u \cdot \theta^{v_i}\right) \right] \cdot e_u. \tag{16}$$

The update formula of the embedding vector $w(u)$ of each word is as follows:

$$w(u) := w(u) + \mu \sum_{v_i \in \{v\} \cup NEG(v)} \left[ L^{v_i}(u) - \sigma\left(e_u \cdot \theta^{v_i}\right) \right] \cdot \theta^{v_i}. \tag{17}$$

with $\mu$ representing the learning rate. After obtaining the updated values of $\theta^{v_i}$ and $e_u$, the objective function can be iteratively optimized. Weight $\alpha$ needs to be multiplied into Eqs. (13) and (14) $\mu$ before balancing the network structure model and text feature model. $e_u$ is the sum of the expression vectors of the text words, and $w(u)$ is the representation vector of the word $u$ in the target node text. The attention function *att* is used to weigh the contribution rate of different text words to the model.

When context words act as context nodes, the sum of context vectors is $e_u$, and its calculation is as follows:

$$e_u = \sum_{j=1}^{|S|} att\left(w_j\right) \cdot d_j. \tag{18}$$

| Table 1 | Dataset description. | | |
|---------|------------------|-------|----------------|
| **Dataset** | **Nodes** | **Edges** | **Average Degree** |
| CiteSeer | 4610 | 5923 | 2.57 |
| DBLP | 17725 | 105781 | 11.926 |
| SDBLP | 3119 | 39516 | 25.339 |

In Eq. (15), $att(w_j)$ is the attention weight of the word $w_j$ to the target node, $d_j$ is the expression vector of the word $w_j$, and $|S|$ is the number of words in the node text.

The attention function $att$ is calculated as follows:

$$att\left(w_j\right) = \frac{\exp(d_j \cdot \mathrm{w}(u))}{\sum_{k=1}^{|S|} \exp(d_j \cdot \mathrm{w}(u))}. \tag{19}$$

The attention weight in Eq. (16) is calculated by the vector of words in the text and the vector of context nodes.

## EXPERIMENTS

### Dataset

To evaluate the effectiveness of the TAMNR model, three citation network datasets are used: the academic network dataset CiteSeer (M10), Data Base systems and Logic Programming (DBLP; V4), and Simplified Data Base systems and Logic Programming (SDBLP). SDBLP is used to remove the nodes with less than three references in the DBLP. The average degree of the CiteSeer (M10) dataset is only 2.57, making it a typical sparse dataset; DBLP (V4) has an average degree of 11.297, making it a relatively dense dataset; and SDBLP has an average degree of 25.337, making it to be a denser dataset. These three datasets are selected to test the model on a variety of dataset sizes and to simulate different types of network types in real life, verifying that the model has good machine learning performance in various networks.

Dataset descriptions can be found in Table 1.

### Introduction to comparison algorithms

DeepWalk: DeepWalk originates from the Word2Vec algorithm. The DeepWalk algorithm is the most classic network embedding learning algorithm based on neural networks. Most subsequent network embedding learning algorithms are based on the DeepWalk algorithm. DeepWalk can use the CBOW model with fast training speed and the Skip-Gram model with high training accuracy to train the representation learning model based on the neural network, and can also use negative sampling and hierarchical SoftMax to accelerate the network training process. In this article, DeepWalk is trained using CBOW and negative sampling.

LINE: LINE is a network representation algorithm that encodes the network structure of a very large-scale network into a low-dimensional network by sacrificing accuracy. Therefore, the training speed of LINE is very fast, but the accuracy is low, especially in sparse networks. LINE's speed improvement comes from only considering the first-order

similarity or second-order similarity of the network, and the concatenated representation of the two.

GraRep: The GraRep algorithm is based on the idea that the high-order similarity between nodes is important in generating the global representation of nodes. The algorithm uses $k$-step similarity and calculates its state transfer matrix for different $k$ values. The algorithm transforms the optimization problem of the loss function into a matrix decomposition problem, and directly obtains the global representation matrix of the graph through SVD decomposition, with each row representing the global representation vector of a node.

MFDW: Because DeepWalk decomposes the matrix $M = (A + A^2)/2$, MFDW uses the SVD algorithm to decompose the matrix M, and uses $W = U \cdot S^{0.5}$ as the embedding vector of the network.

Text Feature (TF): TF converts the text content of the network node into a co-occurrence matrix, and then uses SVD to decompose this co-occurrence matrix to obtain a text feature vector with a column dimension of 100. The TF method is a content-based contrast algorithm.

TADW: TADW is a matrix factorization algorithm that decomposes a matrix $M$ and text matrix T. TADW does not consider context information, and its $T$ matrix cannot preserve the order of words.

CAHNE: CAHNE learns context embeddings for nodes by introducing the context node sequence, and the attention mechanism is also integrated into the model to better reflect the impact of context nodes on the current node.

## Experimental setup

The network node classification task is used to evaluate the algorithm proposed in this article against the comparison algorithms introduced in this article, with Liblinear as the baseline classifier. To verify the generalization ability of the algorithm, the training set is set to 0.1~0.9, and the remaining network nodes are used as the test set. The network embedding vector obtained by the network embedding learning algorithm is uniformly set to 100 dimensions, the random walk length is set to 40, the number of random walks to 10, the window size to 5, the negative sampling to 5, the minimum node frequency to 5, and the learning rate of the neural network is set to 0.05. All experiments in this article are repeated 10 times and then averaged for the final result.

## Analysis of experimental results

Three real network datasets, CiteSeer, DBLP, and SDBLP, are used as evaluation datasets, with 10% to 90% of the dataset used as the training set, and the remaining data used as the test set. Table 2 lists the network node classification accuracy.

As shown in Table 2, the network node classification performance of the LINE algorithm is the worst. MFDW is the matrix factorization form of DeepWalk, and its network node classification performance is better than the DeepWalk algorithm in the training sets of various proportions. On the CiteSeer dataset, the MFDW text features of network nodes outperformed the the DeepWalk algorithm in the node classification task. The TAMNR

**Table 2  Contrast experimental results of node classification task.**

| Dataset | Contrast methods | Dataset percentage (%) | | | | | |
|---|---|---|---|---|---|---|---|
| | | 10% | 20% | 30% | 40% | 50% | 60% |
| CiteSeer | DeepWalk | 55.89 | 59.30 | 60.89 | 61.48 | 62.19 | 62.30 |
| | LINE | 42.64 | 47.06 | 48.04 | 49.57 | 50.43 | 51.02 |
| | GraRep | 39.38 | 53.09 | 57.85 | 59.75 | 59.97 | 61.05 |
| | MFDW | 57.62 | 60.79 | 62.33 | 63.05 | 62.96 | 63.00 |
| | TF | 57.69 | 61.30 | 62.76 | 63.05 | 63.48 | 63.30 |
| | CAHNE | 59.12 | 63.55 | 64.85 | 65.49 | 65.94 | 68.05 |
| | TAMNR | 64.96 | 67.60 | 68.92 | 69.58 | 69.56 | 70.03 |
| DBLP | DeepWalk | 62.26 | 64.34 | 65.42 | 65.98 | 66.24 | 66.18 |
| | LINE | 64.49 | 66.53 | 67.49 | 67.87 | 67.98 | 68.30 |
| | GraRep | 58.92 | 65.92 | 67.26 | 67.92 | 68.77 | 68.88 |
| | MFDW | 65.02 | 74.68 | 74.88 | 75.02 | 75.05 | 75.13 |
| | TF | 66.17 | 69.46 | 70.49 | 71.15 | 71.29 | 71.44 |
| | CAHNE | 65.42 | 67.38 | 68.87 | 69.26 | 71.29 | 72.67 |
| | TAMNR | 71.10 | 72.88 | 73.75 | 73.88 | 74.52 | 74.67 |
| SDBLP | DeepWalk | 79.76 | 80.65 | 81.88 | 81.49 | 82.56 | 82.35 |
| | LINE | 73.79 | 77.01 | 78.11 | 81.49 | 79.31 | 78.97 |
| | GraRep | 80.99 | 82.52 | 84.14 | 84.78 | 84.97 | 84.17 |
| | MFDW | 79.79 | 83.08 | 84.38 | 84.12 | 84.53 | 84.29 |
| | TF | 65.03 | 71.23 | 72.64 | 73.86 | 74.54 | 75.07 |
| | CAHNE | 66.87 | 68.28 | 69.85 | 70.31 | 71.17 | 72.33 |
| | TAMNR | 84.55 | 84.96 | 84.93 | 85.19 | 85.00 | 85.36 |

model proposed in this article adds the node text feature relationship, so its performance is also better than the DeepWalk algorithm.

In the DBLP and SDBLP networks, the network node classification performance of DeepWalk is slightly inferior to that of LINE and GraRep. The MFDW algorithm based on DeepWalk performed better than DeepWalk. As the training set ratio increased, the network node classification performance of Text Feature surpassed DeepWalk. Since DBLP and SDBLP are denser networks, the TAMNR algorithm performed better than the comparison algorithms on these networks.

On the CiteSeer dataset, the average classification accuracy of the TAMNR algorithm is 69.59%, and the average classification accuracy of the DeepWalk algorithm is 61.88%. On the DBLP dataset, the average classification accuracy of the TAMNR algorithm is 73.92%, and the average classification accuracy of the DeepWalk algorithm is 65.64%. On the SDBLP dataset, the average classification accuracy of the TAMNR algorithm is 85.18%, and the average classification accuracy of the DeepWalk algorithm is 81.94%.

The comparison graph of these results is shown, as follows:

As shown in Fig. 2, the network node classification accuracy value span of these algorithms on the CiteSeer and DBLP datasets are larger than on the SDBLP dataset. On the SDBLP dataset, the network node classification accuracy of the six comparison algorithms showed a significant upward trend. However, on the CiteSeer and DBLP

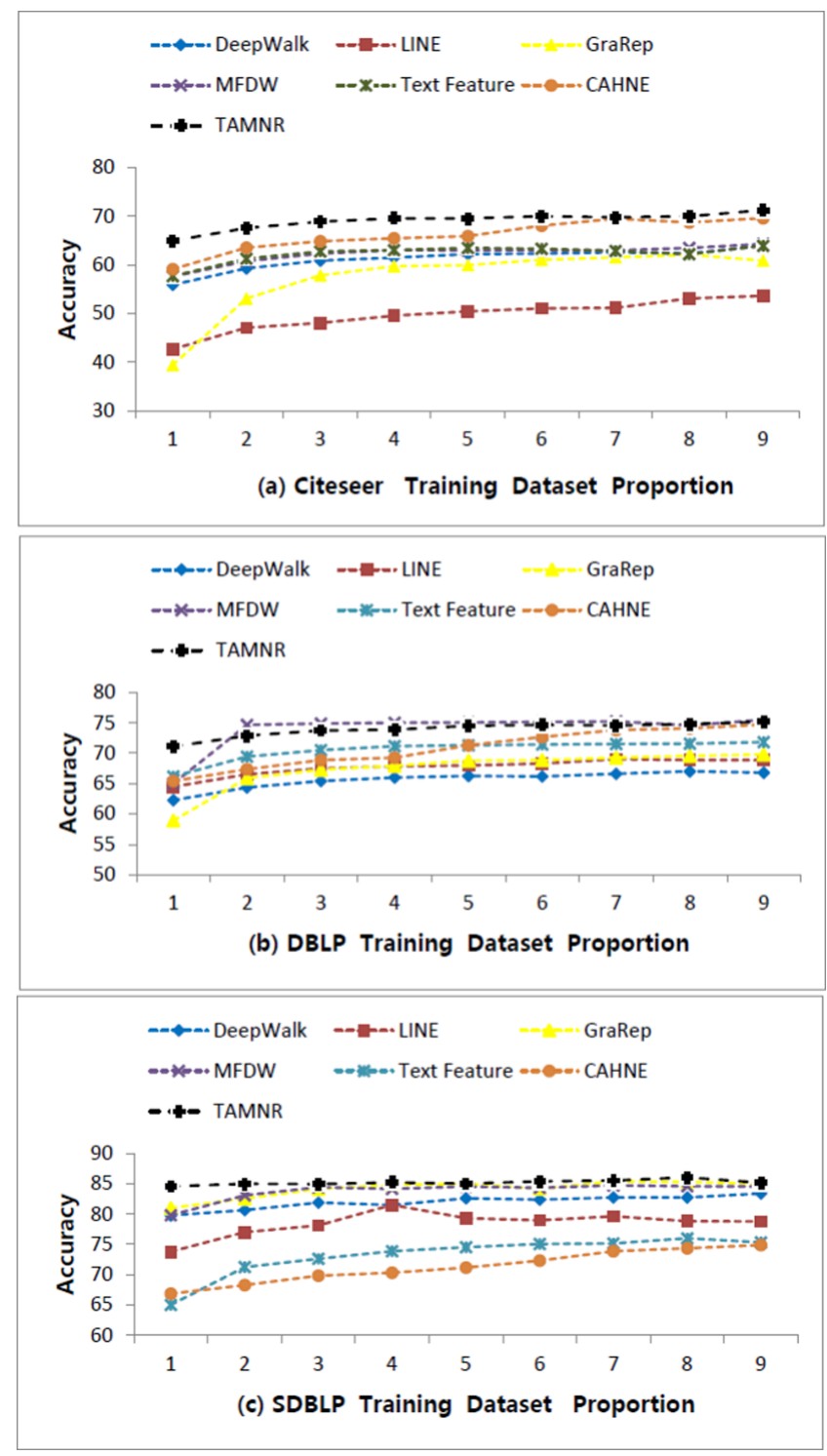

**Figure 2** Performance comparisons of six algorithms on three datasets.

datasets, the accuracy curve showed a relatively slower upward trend. The main reason for this observed difference is that the features obtained by different algorithms are quite different on a sparse network. If an algorithm can obtain features that fully reflect the network structure, its network classification performance is better. On sparse networks, the network representation learning algorithm based on joint learning can make up for the insufficient training caused by the sparse edges. On dense networks, different algorithms can also obtain effective network structure features from sufficient edge connections, so the difference between the classification performance is smaller.

Four main results are obtained from this comparison study: (1) On the CiteSeer sparse network, the classification performance of GraRep based on high-order representation is not as good as that of DeepWalk, but on dense network datasets, such as DBLP and SDBLP, the classification performance of GraRep is better than that of DeepWalk. (2) On dense network datasets, such as DBLP and SDBLP, the network representation learning algorithm based on matrix decomposition is more effective than the network representation learning algorithm based on a shallow neural model, and the former performed slightly better than the latter in the network node classification task. (3) There are many ways to integrate the text features of network nodes. The simple vector concatenate method cannot bring about significant performance improvement in network representation tasks. Induced matrix completion is a very effective text feature integration framework, and it achieved excellent network representation performance on the three real datasets. The text feature integration framework proposed in this article overcomes the computational limitation of matrix decomposition, and uses node text features to constrain the DeepWalk training procedure, so that the network embedding vectors contain more semantic information. (4) On sparse networks, such as CiteSeer, the classification performance of TAMNR model proposed in this article is better than the comparison algorithms, but as the average degree of network nodes increased, this difference in performance became smaller. On the DBLP and SDBLP datasets, the classification performance difference between TAMNR and DeepWalk is small.

## Network embedding visualization

The main purpose of network representation visualization is to check whether the representation vectors obtained by training show a significant clustering phenomenon. The clustering phenomenon shows whether the network representation has learned the community information of the network. If the community division based on the network representation obtained is more accurate, it has better reliability in the network node classification task. In this experiment, four classes of nodes are randomly selected from the CiteSeer dataset, and 150 nodes are randomly selected for each class. The t-SNE algorithm is used to visualize the learned network representation. The results are shown in Fig. 3.

DeepWalk performed poorly in network representation classification tasks, so DeepWalk also displayed the worst results in visualization tasks. The network representation learning performance of the TAMNR model on the three data sets demonstrated its excellent node classification performance, so the embedding vectors obtained by TAMNR showed obvious clustering phenomenon and clustering boundaries. This visualization experiment shows

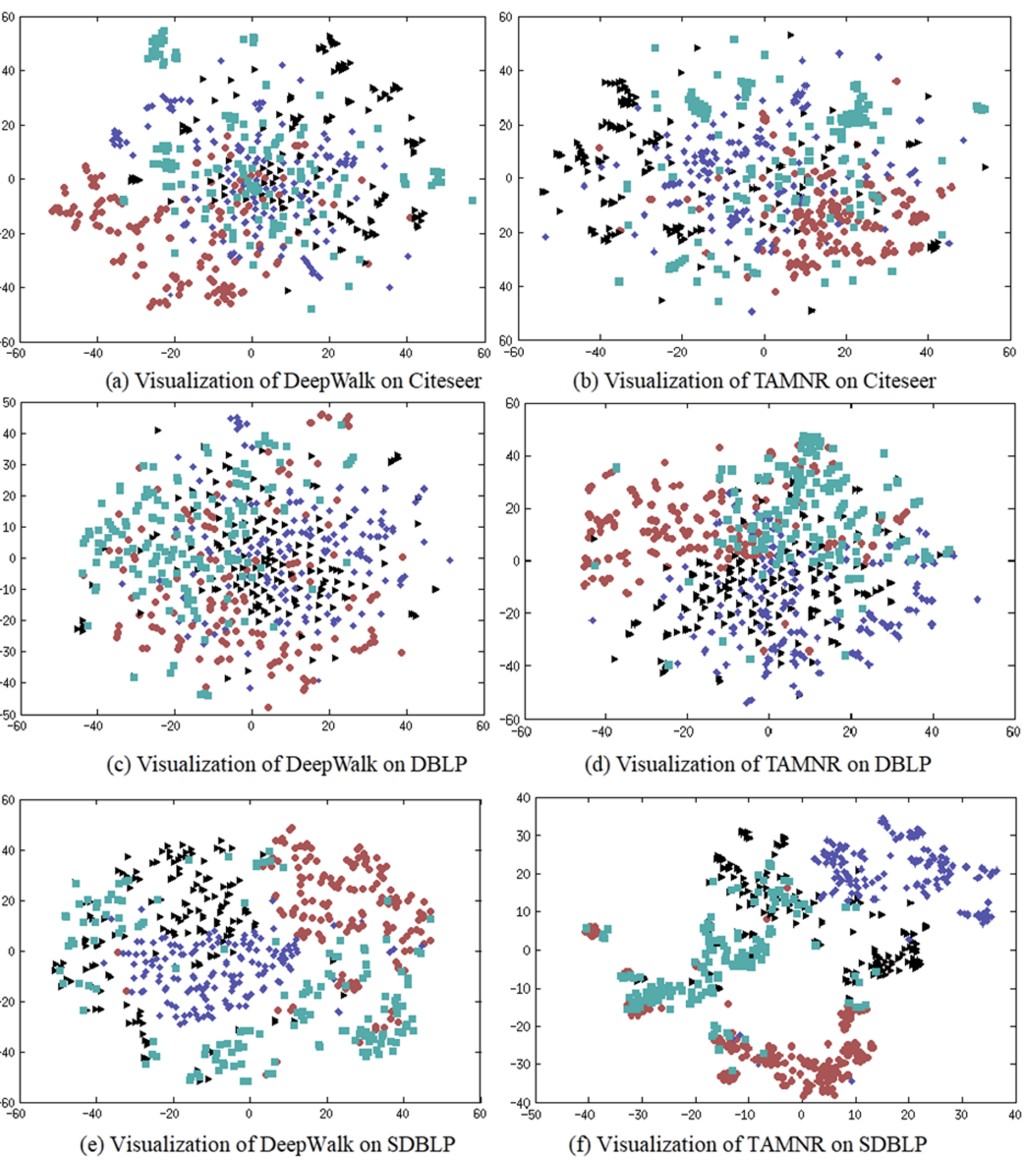

**Figure 3** **(A–F) Network node visualization using DeepWalk and TAMNR.** (A) Visualization of Deep-Walk on Citeseer. (B) Visualization of TAMNR on Citeseer (C) Visualization of DeepWalk on DBLP. (D) Visualization of TAMNR on Citeseer. (E) Visualization of DeepWalk on SDBLP. (F) Visualization of DeepWalk on Citeseer.

that the proposed strategy of combining node text features can improve the performance of network embedding.

## Case analysis

In order to verify the feasibility of the TAMNR model proposed in this article, the target node "Quantum Field Theory as Dynamical System" is randomly selected on the CiteSeer dataset. Three nodes with higher similarity with the target node are obtained by cosine calculation. The result is shown in Table 3.

**Table 3  Case analysis.**

| Algorithm | Vertex Title |
|---|---|
| DeepWalk | On the pct-theorem in the theory of local observables |
| | In the theory of local observables |
| | Modular covariance PCT spin and statistics |
| TADW | Remarks on causality in relativistic quantum field theory |
| | Quantum field theory as eigenvalue |
| | Kam theorem and quantum field theory |
| TAMNR | On the pct-theorem in the theory of local observables |
| | Statistics localization regions and modular symmetries in quantum field theory |
| | Charged sectors spin and statistics in quantum field theory on curved spacetimes |

As shown in Table 3, the DeepWalk model only considers the similarity of network structure features, and does not consider the similarity of text features, so similar nodes did not reflect the text feature similarity. The TAMNR model takes both the network structure features and the text features into account, so the returned similar nodes showed word co-occurrence. Both the TADW algorithm and the TAMNR algorithm proposed in this article consider the text feature information, but the returned nodes are different. The main reason for this difference is that the two algorithms have different mechanisms for modeling text features, so the TADW or TAMNR algorithms are likely best suited to different tasks.

## SUMMARY

This article proposes a new algorithm, called TAMNR, that can encode text information and has a textual attention mechanism added. The goal of this model is to add more information as network features in the modeling procedure. To consider the semantic relationships between nodes, text features in network nodes are introduced in this model. The added attention model ensures that the more important text features have higher weights. In different machine learning tasks, the results of the TAMNR model are significantly better than other comparison models. Adding text features with an attention mechanism can significantly improve the performance of network embedding learning. Future research should focus on how to add new features for joint learning.

### Funding
This research was funded by the National Key R&D Program of China (2020YFC1523300), the Youth Program of Natural Science Foundation of Qinghai Province (2021-ZJ-946Q), the Independent Project on State Key Laboratory of Tibetan Intelligent Information Processing and Application (2022-SKL-001), and the National Natural Science Foundation

of China (No. 11661069, No. 61763041, No. 61663041). The funders had no role in study design, data collection and analysis, decision to publish, or preparation of the manuscript.

### Grant Disclosures

The following grant information was disclosed by the authors:
National Key R&D Program of China: 2020YFC1523300.
Youth Program of Natural Science Foundation of Qinghai Province: 2021-ZJ-946Q.
Independent Project on State Key Laboratory of Tibetan Intelligent Information Processing and Application: 2022-SKL-001.
National Natural Science Foundation of China: 11661069, 61763041, 61663041.

### Competing Interests

The authors declare there are no competing interests. Xiaojuan Ma is employed by Qinghai Provincial Radio and Television Bureau.

### Author Contributions

- Wei Zhang conceived and designed the experiments, performed the experiments, analyzed the data, performed the computation work, prepared figures and/or tables, authored or reviewed drafts of the article, and approved the final draft.
- Zhonglin Ye performed the experiments, authored or reviewed drafts of the article, and approved the final draft.
- Haixing Zhao conceived and designed the experiments, authored or reviewed drafts of the article, and approved the final draft.
- Jingjing Lin analyzed the data, prepared figures and/or tables, and approved the final draft.
- Xiaojuan Ma analyzed the data, prepared figures and/or tables, and approved the final draft.

### Data Availability

  The data is available at GitHub and Zenodo:
  - https://github.com/yezhonglin/TAMNR
  - Wei Zhang, & Zhonglin Ye. (2023). CiteSeer(M10),DBLP(V4),SDBLP. https://doi.org/10.5281/zenodo.8414240

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
