# Peer review of "TAMNR: a network embedding learning algorithm using text attention mechanism"

_PeerJ Computer Science, doi:10.7717/peerj-cs.1736_

## Round 0.1 · original submission · Major Revisions

This paper proposes a network embedding learning algorithm combining network structure and node text features. The contrast experiment in this paper needs to be strengthened, and please check whether the relevant information of the affiliation is correct or not.

**Language Note:** The review process has identified that the English language must be improved. PeerJ can provide language editing services - please contact us at [email protected] for pricing (be sure to provide your manuscript number and title). Alternatively, you should make your own arrangements to improve the language quality and provide details in your response letter. – PeerJ Staff

Reviewer 1 ·

Basic reporting

This paper proposes a network embedding learning algorithm combining network structure and node text features, which adds an attention mechanism to each word in node texts. But,The article has the following problems.

1.The address of the first work school provided by the author is problematic, and the author works for too many institutions, which makes people confused.
2.Abstract only describes the problem and the proposed new method, but does not point out the innovation of the new method and the main framework, the experimental summary is too simple.
3.The author shows that CBOW model is a commonly used model for solving word2vec, and what improvements and innovations have been made in it need to be clearly given and introduced in detail.
4.In introducing TAMNR Modeling, the author needs to show how attention is achieved, and the given figure 1 does not help the reader understand the design idea well.
5.The method proposed in this paper is based on attention mechanism, but there is no such algorithm in the comparison experiment, which cannot show the effectiveness of its algorithm. Please add 1-2 latest related algorithms for comparison experiment.

Experimental design

no comment

Validity of the findings

The method proposed in this paper is based on attention mechanism, but there is no such algorithm in the comparison experiment, which cannot show the effectiveness of its algorithm. Please add 1-2 latest related algorithms for comparison experiment.

Additional comments

no comment

Cite this review as

·

Basic reporting

The logic is good, but the word should be in accordance with the language criterion.

Experimental design

1. Why did you use these three open citation network datasets for testing? The authors should explain more about the selection of the datasets.
2. The article needs more experiments to verify that the new approach can be applied to more domains.

Validity of the findings

The authors need to discussion more about embedding.You can refer to the research article as follows, which might be helpful to you:
Zhang, C., Shang, K., & Qiao, J. (2021). Adaptive Similarity Function with Structural Features of Network Embedding for Missing Link Prediction. Complex., 2021,

---

## Round 0.2 · Minor Revisions

The paper has still some problems as follows:
1. Format problem, text does not have two ends aligned.
2. Reference about deep learning should be cited.

Reviewer 1 ·

Basic reporting

The author basically made revisions according to the comments of the reviewers.
Authors also need to be aware of the following issues.


1.CBOW is a commonly used model, and CBOW model using negative sampling has also been used by some people, which cannot be counted as their own innovation points, but can only be described in related work. The full name of the model should be given when it is first used. If there are innovations, highlight the differences from traditional methods.

2.There are some problems with the narration and the use of words in some sentences. Please check carefully.

Experimental design

no comment

Validity of the findings

no comment

Additional comments

no comment

Cite this review as

·

Basic reporting

It is good enough for this journal.

Experimental design

It is clear and good enough for this journal.

Validity of the findings

I can accept this version. It is good enough for this journal.

Additional comments

None

---

## Round 0.3 · accepted · Accept

The paper has been revised to address all the problems identified in the review process